# The Virome of Babaco (*Vasconcellea × heilbornii*) Expands to Include New Members of the *Rhabdoviridae* and *Bromoviridae*

**DOI:** 10.3390/v15061380

**Published:** 2023-06-16

**Authors:** Edison G. Reyes-Proaño, Maria G. Cañada-Bautista, Juan F. Cornejo-Franco, Robert A. Alvarez-Quinto, Dimitre Mollov, Eduardo Sanchez-Timm, Diego F. Quito-Avila

**Affiliations:** 1Facultad de Ciencias de la Vida, Escuela Superior Politécnica del Litoral, ESPOL, Km 30.5 Vía Perimetral Campus Gustavo Galindo, Guayaquil 090902, Ecuador; edison@uidaho.edu (E.G.R.-P.); mcanada@espol.edu.ec (M.G.C.-B.); lesanche@espol.edu.ec (E.S.-T.); 2Department of Entomology and Plant Pathology, University of Idaho, Moscow, ID 83843, USA; 3Centro de Investigaciones Biotecnológicas del Ecuador, Escuela Superior Politécnica del Litoral, CIBE-ESPOL, Km 30.5 Vía Perimetral Campus Gustavo Galindo, Guayaquil 090902, Ecuador; jcornejo@espol.edu.ec; 4Department of Botany and Plant Pathology, Oregon State University, Corvallis, OR 97330, USA; alvarero@oregonstate.edu; 5USDA-ARS, Corvallis, OR 97330, USA; dimitre.mollov@usda.gov

**Keywords:** *Nucleorhabdovirus*, *Ilarvirus*, high-throughput sequencing, plant virus discovery, mountain papaya

## Abstract

Babaco (*Vasconcellea × heilbornii*) is a subtropical species in the *Caricaceae* family. The plant is native to Ecuador and represents an important crop for hundreds of families. The objective of this study was to characterize, at the genomic level, two new babaco viruses identified by high-throughput sequencing. The viruses, an ilarvirus and a nucleorhabdovirus, were found in a symptomatic babaco plant from a commercial nursery in the Azuay province of Ecuador. The tripartite genome of the new ilarvirus, provisionally named babaco ilarvirus 1 (BabIV-1), is related to subgroup 3 ilarviruses, including apple mosaic virus, apple necrotic mosaic virus, and prunus necrotic ringspot virus as the closest relatives. The genome of the nucleorhabdovirus, provisionally named babaco nucleorhabdovirus 1 (BabRV-1), showed the closest relation with joa yellow blotch-associated virus and potato yellow dwarf nucleorhabdovirus. Molecular-based detection methods found BabIV-1 and BabRV-1 in 21% and 36%, respectively, of plants surveyed in a commercial babaco nursery, highlighting the importance of enforcing virus testing and nursery certification programs for babaco.

## 1. Introduction

Babaco (*Vasconcellea × heilbornii*) is an herbaceous tree native to the Ecuadorian highlands. It belongs to the *Caricaeae* family and shares morphological and anatomical traits with its tropical relative, papaya (*Carica papaya*) [1]. The babaco fruit has unique and exotic organoleptic properties and has been commercialized fresh or processed in local and international markets. In Ecuador, babaco is produced in small orchards, usually under greenhouse conditions. Nevertheless, the vegetative propagation, the lack of pathogen-free certified material, as well as the unregulated nursery operations have caused an exponential dissemination of viruses in the main producing areas, threatening the profitability and sustainability of this native crop [2,3].

In the last decade, high-throughput sequencing (HTS) has advanced the discovery of pathogenic and non-pathogenic viruses in agricultural and ornamental ecosystems [4]. In babaco, HTS has identified several disease-associated viruses including babaco mosaic virus (BabMV, genus *Potexvirus*), papaya ringspot virus (PRSV, genus Potyvirus), babaco virus Q (BabVQ, umbra-like virus), babaco cheravirus 1 (BabChV-1, genus *Cheravirus*), and babaco nepovirus 1 (BabNV-1, genus *Nepovirus*) [2,3]. In addition, the genomic sequence for putative cryptic viruses, such as babaco cryptic virus-1 (BabCV-1, a partiti-like virus) and babaco endogenous pararetrovirus (BabEPV, a hypothetical integrated pararetrovirus), has been reported [3].

Here, we report two additional babaco-infecting viruses belonging to distinct genera, *Ilarvirus* (family *Bromoviridae*) and *Alphanucleorhabdovirus* (family *Rhabdoviridae*). The genus Ilarvirus currently comprises 22 species recognized by the International Committee on Taxonomy of Viruses [5]. Ilarviruses have non-enveloped, spherical virions that are 16–36 nm in diameter and single-stranded positive-sense tripartite RNA genome [6]. RNA 1 contains a single open reading frame (ORF) coding for the virus replicase (Protein 1a). Depending on the species, RNA 2 can have one or two ORFs encoding a viral polymerase (ORF 2a) and an RNA silencing suppressor (ORF 2b), expressed by a subgenomic RNA [7,8]. RNA 3 contains two ORFs (ORF 3a and 3b) which code for the movement protein (MP) and the coat protein (CP), respectively [8]. Ilarviruses are pollen- and seed-borne but can also be plant-to-plant transmitted by thrips, which act as carriers of virus-contaminated pollen [9].

The family *Rhabdoviridae* is a large and diverse family of viruses with unsegmented or bi-segmented negative-sense single-stranded RNA genomes that infect vertebrates, invertebrates, and plants. Rhabdovirus virions typically consist of bullet-shaped or bacilliform particles ranging from 100 to 430 nm in length and 45 to 100 nm in diameter, whereas some consist of non-enveloped filamentous particles [10].

The subfamily Betarhabdovirinae groups six genera of plant- and arthropod-infecting rhabdoviruses: Alphanucleorhabdovirus, Betanucleorhabdovirus, Gammanucleorhabdovirus, Cytorhabdovirus, Dichorhavirus, and Varicosavirus. Alphanucleorhabdoviruses, having unsegmented genomes of 11–15 kb, have been reported from monocots and dicots and are transmitted in a persistent-replicative manner by leafhoppers, planthoppers, or aphids [11,12]. The genome organization of alphanucleorhabdoviruses includes leader (l) and trailer (t) sequences at the 3′ and 5′ of the genome, respectively, and enclose six canonical ORFs encoding the nucleocapsid (N), phosphoprotein (P), the movement protein (MP), also referred to as Y protein, the matrix (M) protein, the glycoprotein (G), and the polymerase (L) [13,14].

The objective of this study was to characterize, at the genomic level, two new babaco viruses and infer their genetic relationships with other members of the *Bromoviridae* and *Rhabdoviridae* families, respectively.

## 2. Materials and Methods

### 2.1. Plant Material and Virus Detection

A single six-month-old babaco plant showing virus-like symptoms, such as leaf yellowing, premature leaf abscission, and mild leaf mosaic (Figure 1), was obtained from a commercial nursery in the Azuay province in Ecuador and was selected for this study. The plant was kept under greenhouse conditions and tested by reverse-transcription PCR (RT-PCR) for a panel of known babaco-infecting viruses described so far [3], including BabMV, PRSV, BabChV-1, BabNV-1, BabVQ, BabCV, and BabEPV, using the primers listed in Appendix A.

Total RNA was extracted following a modified protocol described by Halgren, A. et al. [15]. Briefly, 100 mg of leaf tissue were ground and homogenized in 1 mL of lithium-chloride based RNA-extraction buffer containing 1% β-mercaptoethanol. The supernatant was mixed with an equal volume of 6 M potassium-acetate (pH 6.5) and centrifuged; then, the resulting supernatant was mixed (1:1) with isopropanol and centrifuged to form a pellet. The pellet was precipitated by centrifugation and rinsed two times with wash buffer (TrisHCl + NaCl + EDTA) + Ethanol (1:1) in the presence of glassmilk. Finally, the glassmilk–RNA complex was vacuum dried, and the RNA was resuspended in 100 μL of water and stored at −20 °C.

Reverse transcription (RT) was carried out in a 15 µL reaction containing 0.1 µL of SuperScript III (Thermo Fisher, Waltham, MA, USA), 0.5 µL of random hexamers (40 µM), and 1 µL of RNA template (~45 ng) following manufacturer instructions. PCR was completed using GoTaq^®^ Green Master mix (Promega, Madison, WI, USA) in 10 µL reactions with 1 µL of template and 0.5 µL of each primer at 40 µM. To validate RNA quality and synthesized cDNA, a 721 nt region corresponding to NADH dehydrogenase ND2-subunit was amplified as a housekeeping gene by primers F: 5′-GGACTCCTGACGTATACGAAGGATC-3′ and R: 5′-AGTAGATGCTATCACACATACAAT-3′ [16,17]. To detect the presence of BabIV-1 and BabRV-1, specific primers were designed. For BabIV-1 detection, a 397 nt fragment corresponding to a fragment of the MP and the CP was amplified by F: 5′-GGTTAGAAGCAGGATTGCGTC-3′ and R: 5′-CTTGAGCTCCAGTCGGATGAC-3′. On the other hand, primers F: 5′-GGAGTCGGCATTGGTATAGCA-3′ and R: 5′-CCGTTGTCCTTTAACACCCTAG-3′ targeted a 923 nt region corresponding to the N and P genes from BabRV-1. PCR conditions were as follows: initial denaturation at 95 °C for 4 min followed by 40 cycles of 95 °C for 40 s, 58 °C for 40 s, and 1 min at 72 °C, and a final extension step at 72 °C for 7 min.

### 2.2. Virus Sequencing, Assembly, and Annotation

The nearly complete genome was obtained by HTS from total RNA extracted from 100 mg of symptomatic leaf tissue using the RNeasy Plant Mini Kit (Qiagen, Hilden, North Rhine-Westphalia, Germany). The total RNA was subjected to ribosomal-RNA removal by using the Ribo-Zero Plant kit. The cDNA library was prepared by Illumina’s TruSeq Stranded total RNA and sequenced on an Illumina HiSeq 4000 platform as pair-ended reads (2 × 150 bp) at Macrogen Inc. (Teheran-ro, Gangnam-gu, Seoul, Republic of Korea). Raw reads were preprocessed using the BBDuk adapter/quality trimming tool version 38.84 to remove adapters and low-quality reads. Trimmed reads were assembled using SPAdes assembler version 3.15.2 [18]. Since the genome of babaco has not been determined thus far, assembled contigs were mapped to the genome of *Carica papaya* (NCBI PRJNA264084) to filter out common contigs between these two members of the *Caricaceae* family. Non-host contigs were analyzed by BLAST search using a custom viral sequences database (retrieved from the NCBI in November 2021) and the full NCBI non-redundant protein sequences (nr) database.

Putative viral genomes were confirmed by Sanger sequencing of overlapping RT-PCR products using the primers listed in Appendix A. Amplicons were generated using the proof-reading polymerase Platinum SuperFi II DNA (Thermo Fisher, Waltham, MA, USA). RNA termini for each viral genome were determined from double-stranded RNA (dsRNA) using a 5′/3′ RACE Kit, Second Generation System (Roche, Basel, Switzertland), and the primers listed in Appendix A. DsRNA was extracted from fresh leaf tissue following the protocol described by Dodds et al. [19]. Briefly, 15 g of leaf tissue were ground in liquid nitrogen and homogenized for 30 min in a mixture containing 40 mL of 2X STE (50 mM Tris–Cl, 100 mM NaCl, 1 mM EDTA, pH 7.1), 8 mL of 10% SDS, 23 mL of STE-saturated phenol, and 0.5 mL of B-mercaptoethanol. The homogenate was centrifuged at 10,000× *g* for 20 min. The supernatant was transferred to a new flask, measured, and mixed with 0.2 volumes of 100% ethanol and 1.5 g of cellulose fibers medium (Sigma-Aldrich, St. Louis, MO, USA). The cellulose-bound dsRNA was pelleted and washed twice in STE/18%EtOH (18% ethanol, *v*/*v*, in STE). The dsRNA was eluted in 15 mL of STE. DNA and ssRNA were digested by the addition of 30 U of DNAse I (Sigma), 1000 U RNAse T1 (Sigma), and 300 µL of 2 M MgCl_2_ at 37 °C for 1 h. The digestion was terminated by the addition of 1 mL of 0.5 M EDTA, pH 8.0, and 5 mL of 100% EtOH. The dsRNA was homogenized in the presence of 0.2 g of cellulose fibers medium and was precipitated and washed in 30 mL of STE/18%EtOH. The final elution was completed in 3 mL of STE.

Amplified DNA products were cloned using a pGEM^®^-T easy vector system (Promega, Madison, WI, USA) and sequenced at Macrogen Inc. (Teheran-ro, Gangnam-gu, Seoul, Republic of Korea). The complete viral genomes, including the corresponding terminal sequences, were assembled using Geneious Prime 2022.0.2. For genome annotation, putative ORFs were predicted using Unipro UGENE v40.03 [20] and ORFfinder (NCBI). The hypothetical protein homologs were identified by BLASTp and the non-redundant protein sequences (nr) database (https://blast.ncbi.nlm.nih.gov/, accessed on 13 January 2022). Predicted amino acid sequences were analyzed to identify functional conserved domains. This analysis was performed using the Conserved Domains database at NCBI (https://www.ncbi.nlm.nih.gov/Structure/cdd/wrpsb.cgi, accessed on 13 January 2022).

### 2.3. Phylogenetic Analysis

Phylogenetic analyses were performed based on nucleotide and amino acid sequences retrieved from NCBI GenBank and aligned using MUSCLE version 3.8 [21]. The best protein substitution model was inferred using MEGAX [22]. For the new ilarvirus, the analysis was performed for all four predicted proteins encoded by the genome. The first two proteins were related to virus replication (P1a and P2a), movement protein (MP), and coat protein (CP). The best protein models were LG + G + I, rtREV + G + F + I, LG + G, and LG + G + F for P1a, P2a, MP, and CP, respectively. For the rhabdovirus, the amino acid sequence of the viral replicase (L protein) was used, implementing rtREV substitution model. Phylogenetic relationships were estimated using RAxML [23] on CIPRES with 1000 bootstrap replicates.

### 2.4. Mechanical Transmission and Virus Survey

Leaves of the original virus-infected plant were ground in 0.05 M phosphate buffer (pH 7.0) 1:10 (wt/vol), as described by Hull [24]. The two youngest, fully developed leaves of each test plant were dusted with carborundum (silicon carbide 600 mesh) and rubbed with the infectious sap using a soft sponge. Three plant species belonging to the family *Caricaeae,* babaco (*Vasconcellea × heilbornii)*, chamburo (*Vestipulata pubescens*), and papaya (*Carica papaya*), were evaluated for mechanical transmission, with five replicates for babaco and papaya, while 10 chamburo plants were used. Inoculated plants were kept in an insect-proof cage within a greenhouse at 22–25 °C and 12 h natural light. Inoculated plants were virus tested at 15-, 30-, and 90-days post inoculation (dpi).

A group of fifty six-month-old babaco nursery plants showing mild leaf mosaic were sampled for determining the prevalence of the new viruses in a single commercial nursery in the Azuay province. These samples were subjected to total RNA extraction, as described above, and RT-PCR for virus detection using the primers listed in Appendix A.

## 3. Results

### 3.1. Virus Detection

The symptomatic babaco sample investigated in this study tested negative by RT-PCR for BabMV, PRSV, BabChV-1, and BabNV-1, which have been shown to be associated with leaf symptoms in babaco [3]. BabVQ, an umbra-like virus considered non-pathogenic [25], along with BabCV and BabEPV, both asymptomatic ‘cryptic’ viruses, were RT-PCR detected in the sample.

### 3.2. High-Throughput Sequencing

HTS resulted in a raw dataset of 54,785,178 paired-end reads. Post-trimming reads (54,689,746) were de novo assembled, resulting in 94,378 contigs, from which 23,934 contigs mapped to the genome of papaya (*Carica papaya)*. BLAST search of the remaining contigs (70,444) identified a 4.7 kb-long contig showing 98% identity (100% coverage) to BabVQ. A total of 11 contigs ranging from 0.4 to 2.6 kb in length were assembled from 0.000006% of the total reads. These contigs showed homology to available partial sequences of both BabCV (98% identity) and BabEPV (89% identity), supporting the results obtained by RT-PCR.

Additional contigs of 3.2, 2.5, and 1.7 kb in length were identified with homology (nt identities in the range 67–71%) to corresponding RNA genomic segments of members of the genus *Ilarvirus* (family *Bromoviridae*), whereas a 12.7 kb contig showed homology to members of the genus *Alphanucleorhabdovirus* (family *Rhabdoviridae*). A summary of the most significant virus-related contigs assembled in this study, assembly details, and their corresponding homologous viruses are presented in Table 1. A significant difference in virus-reads abundance was observed between BabVQ (11.94% of total reads) and other viruses (0.006–0.028% of total reads) (Table 1).

### 3.3. Genome Organization, Sequence Comparison, and Phylogenetics of a New Ilarvirus

The complete genomic sequence of a putative new ilarvirus, provisionally named babaco ilarvirus 1 (BabIV-1), consisted of three RNA segments of 3299 nt (RNA1), 2675 nt (RNA2), and 1802 nt (RNA3). The genomic sequence segments of BabIV-1 were deposited in the GenBank under accession numbers OQ256238, OQ256239, and OQ256240, respectively.

RNA 1 contains a single ORF at nucleotide position 33–3143 coding for the hypothetical protein 1a (P1a) of 1037 aa and a predicted molecular weight of 117 kDa. This protein has two conserved domains: the viral methyltransferase (MET, PF01660, F53-T387) and the helicase (HEL, PF01443, K744-T1001) (Figure 2). At the amino acid level, P1a of BabIV-1 is more closely related to that of lilac leaf chlorosis virus (LLCV, YP_009104367.1; 65% identity; 99% coverage) and prunus necrotic ringspot virus (PNRSV, AEP04410; 63% identity; 99% coverage). The motif (_747_DGVAXCXXXT_756_), a common feature in P1a of ilarviruses [26], was identified in the counterpart from BabIV-1.

The single ORF found in RNA 2 is positioned at nucleotides 25–2553 and codes for an 843 aa protein (P2a) of 97 kDa with the conserved viral RNA-dependent RNA polymerase (RdRp) domain (PF00978, T305-R720) (Figure 2). The putative P2a of BabIV-1 shares the highest level of amino acid identity with its counterpart from apple necrotic mosaic virus (ApNMV, AVI05085; 62% identity; 95% coverage). The highly conserved motif _609_ASGDDSLI_616_, which is present in P2a of other ilarviruses [27], was identified within the corresponding P2a of BabIV-1.

RNA 3 contains two non-overlapping ORFs, ORF3a, and ORF3b. Translation of ORF3a (position 96–947 nt) results in a 284 aa protein of 32 kDa having conserved domains of the bromovirus MP family (PF01573, E16-Q229). The amino acid sequence of BabIV-1, the hypothetical MP, showed the highest identity with its homologue from LLCV (YP_009104372.1; 69% identity; 93% coverage). ORF 3b (position 953–1687 nt) putative protein is translated through a subgenomic RNA, producing a 245 aa long protein (P3b) of 27 kDa, with conserved domains present in bromovirus CPs (PF01787, G41-E239) (Figure 2). The predicted CP of BabIV-1 is most closely related to its counterpart from LLCV (YP_009104373.1; 58% identity; 82% coverage) and contains the zinc-finger motif _32_CRLCNHTHAGGCARCKKC_49_. This conserved sequence is also present in the CPs of ilarviruses [8].

Complete genome analysis between BabIV-1 and ilarviruses currently listed in the ICTV revealed average nucleotide identities of 58%, 50%, and 43% for RNA segments 1, 2, and 3, respectively, with species in the ilarvirus subgroup 3. However, the identities between BabIV-1 and ilarviruses belonging to subgroups 1, 2, 4, and unclassified were in the range of 24%–40% for all genomic segments (Appendix A). Consistent with the relationships at the nucleotide level, the highest identities at the amino acid level were observed between predicted proteins of BabIL-1 and counterparts from subgroup 3 ilarviruses (Table 2).

Phylogenetic analyses using the amino acid sequence of each protein showed that BabIV-1 clusters with PNRSV, ApNMV, apple mosaic virus (ApMV), blueberry shock virus (BlShV), and LLCV all belong to ilarvirus subgroup 3 (Figure 3).

### 3.4. Genome Organization, Sequence Comparison, and Phylogenetics of a New Alphanucleorhabdovirus

The genome of the new rhabdovirus, tentatively named babaco rhabdovirus-1 (BabRV-1), was deposited in NCBI GenBank under accession number OQ256237. The single-stranded negative sense monopartite genome of BabRV-1 is 12,802 nucleotides long. Seven ORFs were identified in the genome of BabRV-1. ORF 1 (nt positions: 156–1571) encodes a putative 51 kDa protein containing the conserved rhabdovirus nucleocapsid (N) domain (PF03216: 4.92e-23; K75-I345) with a 71.52% identity to potato yellow dwarf virus (PYDV). ORF 2 (nt positions: 1645–1953) codes for a small 12 kDa protein of unknown function with homology (32.69% aa identity) to protein X, encoded by ORF 2 of physostegia chlorotic mottle virus (PhCMoV). ORF 3 (nt positions: 2013–2852) encodes a putative 31 kDa protein with homology to the phosphoprotein (P) of several members of the *Rhabdoviridae* and shares a 58.06% identity at the amino acid level with joa yellow blotch-associated virus (JYBaV). ORF 4 (nt positions: 2915–3772) encodes a putative 32 kDa protein that contains the highly conserved motif _46_IXD(X)_71_G belonging to the 30K superfamily of viral movement proteins (MP, Interpro IPR041344), while ORF 5 (nt positions: 3974–4747) codes for a putative matrix protein (M) of 29 kDa with a 68.87% of identity to the M protein of PYDV (Table 3).

The last two ORFs code for a putative glycoprotein (G) (ORF 6: nt positions: 4829–6706) with a MW of 69 kDa and the large (L) replicase (ORF 7: nt positions: 6867–12,704), with a MW of 221 kDa; these proteins share a 66.40% and 74.36% identity to PYDV and JYBaV, respectively. The latter has two conserved domains, the RdRp (PF00946: 5.58e-155; W217-L1077) and a mRNA-capping region V (PF14318: 1.34e-43; C1092- E1319), which include an essential conserved motif for mRNA capping _1160_GXXT(X)_72_HR_1235_ [28].

The genome organization of BabRV-1, except for the accessory gene X, which is also observed in the genome of its closest relatives, adheres to the overall organization of nucleorhabdoviruses 3′–N–X–P–MP–M–G–L–5′ (Figure 4). A typical genomic feature observed in rhabdoviruses and other viruses in the order *Mononegavirales* is the presence of conserved intergenic sequences that regulate transcription [29]. In BabRV-1, a highly conserved region was identified in all the gene junctions having the consensus motif 3′-AAUUAUUUUUGGGUUG-5′ (Figure 4), which is also found in the intergenic regions of JYBaV and PYDV. Lastly, the coding region of BabRV-1 is flanked by 3′ leader (155 nt) and 5′ trailer (98 nt) stretches, which are partially complementary to each other (Figure 4).

Phylogenetic inferences based on the amino acid sequence alignment of the L protein of selected members of different genera of the *Rhabdoviridae* family clustered BabRV-1 with members of the genus *Alphanucleorhabdovirus*, with JYBaV and the potato yellow dwarf virus (PYDV) as closest relatives (Figure 5). Furthermore, BabRV-1, JYBaV, and PYDV share an identical genome organization, including the hypothetical X protein gene.

Thus far, the genus *Alphanucleorhabdovirus* includes 12 accepted virus species. The species demarcation criteria for viruses assigned to this genus state three important characteristics: (i) nucleotide sequence identity of complete genomes lower than 75%; (ii) different ecological niches as evidenced by differences in hosts and/or arthropod vectors; and (iii) ability to be distinguished in serological tests or by nucleic acid hybridization [30]. BabRV-1 satisfies two out of the three criteria, sharing 68% nucleotide identity with its closest relatives and having a natural host, babaco, different from those of PYDV and JYBaV, which have been reported in solanaceaeous hosts. Hence, BabRV-1 should be considered a new species within the genus *Alphanucleorhabdovirus*.

### 3.5. Mechanical Inoculation and Virus Survey

Mechanically inoculated plants were tested for BabIV-1 and BabRV-1 and monitored for 90 days. BabIV-1 and BabRV-1 failed to be mechanically inoculated onto papaya, babaco, or chamburo. In babaco plants (*n* = 50) obtained from the commercial nursery in the Azuay province, BabIV-1 and BabRV-1 were detected in 21% and 36%, respectively, while 7% tested positive for both viruses. BabVQ, BabCV, and BabEPV were also detected in all the plants tested.

## 4. Discussion

In the last decade, HTS has advanced the discovery of pathogenic and non-pathogenic viruses in several crop plants around the globe. In vegetatively propagated plants, such as babaco, the identification of viruses constitutes an essential step for sustainable production. Despite previous reports on the occurrence of different viruses in babaco in Ecuador [2,3], a certification scheme in babaco nurseries that prevents the spread of virus-infected material into new production areas has been neglected.

In this study, two new viruses were discovered in babaco plants obtained from a commercial nursery. The viruses, BabIV-1 and BabRV-1, belong to the *Bromoviridae* and *Rhabdoviridae* families, respectively. BabIV-1 has genomic features consistent with those of ilarviruses from subgroup 3 [31]; however, the sequence identity with closest relatives (Table 2) suggests that it should be considered a new member of the genus *Ilarvirus*. BabRV-1 possesses genomic features, including the presence of an accessory gene ‘X’ of unknown function located between the N and P genes, identical to JYBaV, PYDV, PhCMoV, and eggplant mottled dwarf virus (EMDV), which are members of the *Alphanucleorhabdovirus*. The amino acid sequence identity between BabRV-1 and closest relatives (Table 3) suggests its designation as a distinct new member of this genus.

BabIV-1 and BabRV-1 were found in co-infection with BabVQ, an umbra-like virus considered non-pathogenic [25], and two viruses, BabCV and BabEPV, previously regarded as ‘cryptic’ due to the lack of symptoms associated with them [3]. It remains to be determined whether the symptomatology observed in the co-infected plant could be attributed to either BabIV-1, BabRV-1, their interaction with each other, with BabVQ, or with additional viruses not detected in this study.

Some plant rhabdoviruses have been reported either as non-pathogenic or able to cause mild symptoms in single infections [12,32,33], whereas ilarviruses have long been associated with an array of leaf or fruit symptoms, including mosaic, vein banding, ringspot, and necrosis, among others [34,35,36,37]. A particular symptom, commonly known as ‘necrotic shock’, caused by the ilarviruses blueberry shock virus (BlShV) and strawberry necrotic shock virus (SNSV), consists of severe leaf necrosis followed by defoliation [36,38]. Although blueberry and strawberry plants infected with BlShV and SNSV, respectively, recover after the first year of symptoms, important yield losses can be produced during the infection season, and the plants remain infectious for further virus spread in the field [38]. Babaco plants infected with BabIV-1 and BabRV-1 showed premature leaf abscission (Figure 1) followed by an apparent recovery in a pattern similar to the ‘shock’ disease caused by BlShV in blueberry and SNSV in strawberry. In a survey conducted in babaco plants showing mild leaf mosaic in the same nursery where BabIV-1 and BabRV-1 were first identified, the viruses were found in 21% and 36%, respectively, with 7% of plants being positive for both. Premature leaf abscission was not observed at the time of sampling. Hence, pathogenicity studies using infectious clones are warranted to determine the role of each virus (BabIV-1 and BabRV-1) in single and mix infections in symptom expression in babaco and other model hosts.

As for transmission, being a member of the ilarvirus genus, BabIV-1 is most likely transmitted by pollen. In babaco, however, pollination is rare because fruit set occurs by parthenocarpy [39]. BabRV-1, a nucleorhabdovirus, could be transmitted by arthropod vectors, including leafhoppers, planthoppers, or aphids [12]. During our study, we were unable to establish either leafhopper or aphid colonies onto babaco plants to conduct transmission experiments; hence, the natural vector of BabRV-1 remains to be determined. Nevertheless, in commercial operations, the vegetative propagation nature of babaco is probably more efficient at virus dissemination than vector-mediated transmission.

To the best of our knowledge, this is the first report of an ilarvirus and a nucleorhabdovirus infecting babaco. Our findings contribute to the expanding virome of babaco and highlight the importance of virus discovery studies, which are a pivotal component for the establishment of virus testing and nursery certification programs for vegetatively propagated crops.

## Figures and Tables

**Figure 1 viruses-15-01380-f001:**
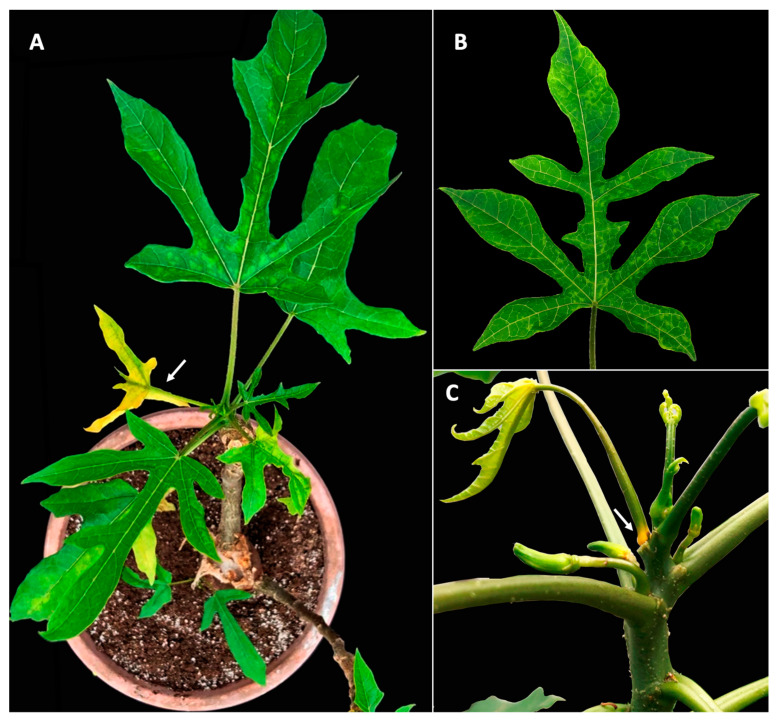
Symptomatic babaco (*Vasconcellea × heilbornii)* plant subjected to this study. (**A**) Leaf mosaic and yellowing (arrow). (**B**) Detached leaf with vein banding and mosaic. (**C**) Yellowing (arrow) at the petiole base of a leaf with premature abscission. Pictures correspond to the same plant at different times.

**Figure 2 viruses-15-01380-f002:**
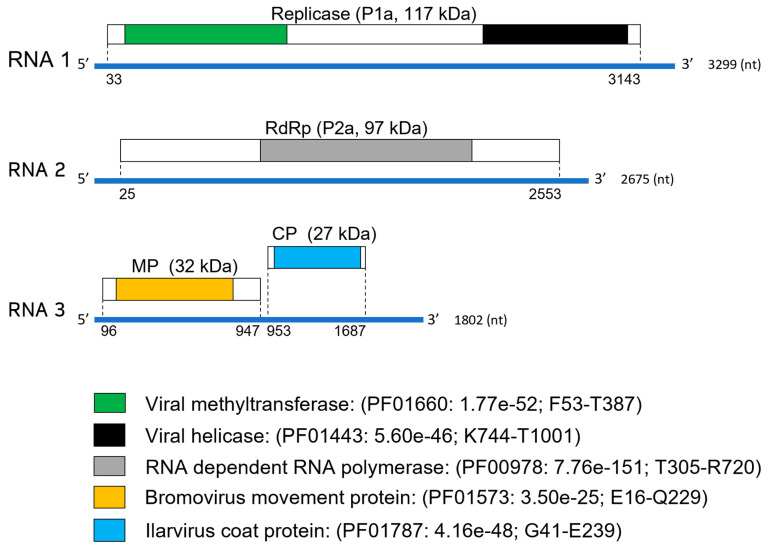
Genome organization of babaco ilarvirus-1. Genome positions of predicted open reading frames (ORFs) in each RNA segment are indicated with dotted lines. Conserved domains and molecular weight (in kDa) of the putative replicase, RNA-dependent-RNA-polymerase (RdRp), movement protein (MP), and coat protein (CP), respectively, are shown in different colors.

**Figure 3 viruses-15-01380-f003:**
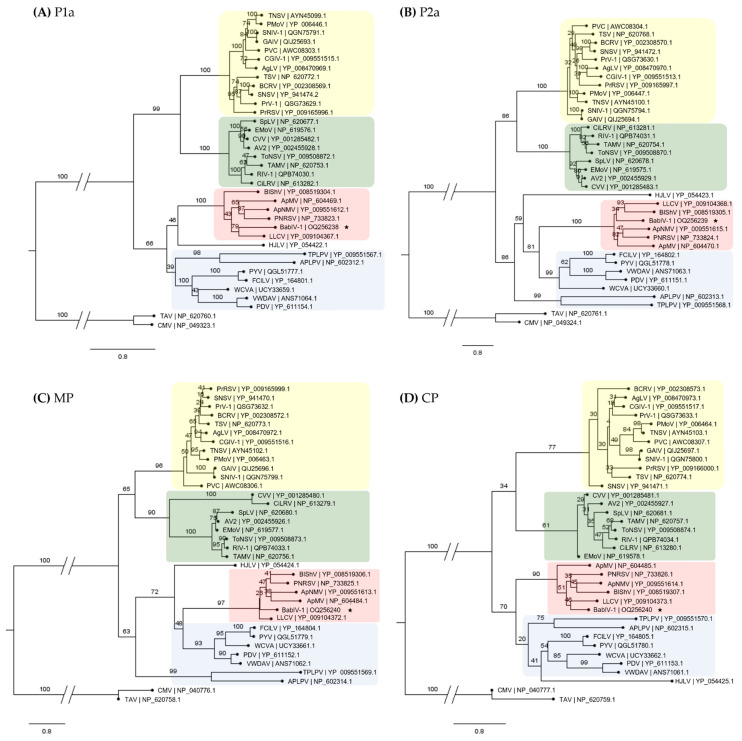
Phylogenetic analysis of babaco ilarvirus-1 (BabIV-1, GenBank accessions: OQ256238, OQ256239, OQ256240) and other ilarviruses based on amino acid sequences of the four proteins, P1a (**A**), P2a (**B**), movement protein (MP) (**C**) and coat protein (CP) (**D**). Colored shading denotes different clades which are consistent regardless of the protein analyzed. The analysis was performed on CIPRES with 1000 bootstraps. Bootstrap values are indicated at each node. Virus abbreviations, names and accession numbers are listed in Appendix A. *Tomato aspermy virus* (TAV) and *Cucumber mosaic virus* (CMV) were used as outgroups. Scale bar represents aminoacids substitution per site. BabIV-1 is marked with an asterisk.

**Figure 4 viruses-15-01380-f004:**
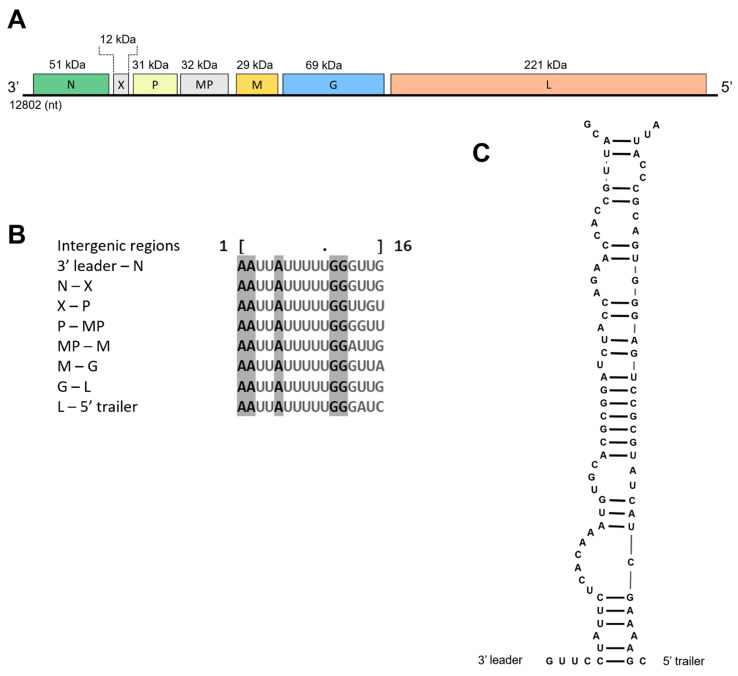
Genomic organization of babaco rhabdovirus-1 (BabRV-1). (**A**) Schematic representation of its predicted open reading frames (ORFs: nucleocapsid (N), protein X (X), phosphoprotein (P), movement protein (MP), matrix (M), glycoprotein (G), polymerase (L)). (**B**) Alignment of intergenic regions observed in BabRV-1 genome. (**C**) Partial genome complementarity of 3′ leader and 5′ trailer terminal sequences.

**Figure 5 viruses-15-01380-f005:**
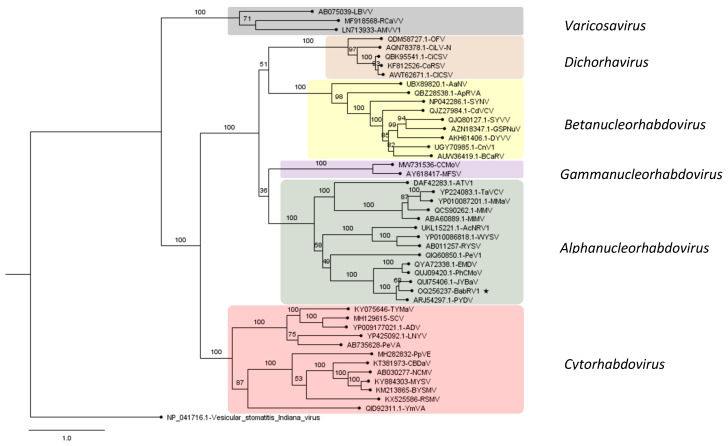
Phylogenetic analysis of babaco nucleorhabdovirus-1 (BabRV-1, GenBank accession: OQ256237) and other rhabdoviruses belonging to subfamily *Betarhabdovirinae*. The analysis was performed on CIPRES with 1000 bootstraps. Bootstrap values are indicated at each node. Virus abbreviations, names, and sequences are listed in Appendix A. *Vesicular stomatitis Indiana virus* was used as an outgroup. Scale bar represents amino acid substitution per site. BabRV-1 is marked with an asterisk.

**Table 1 viruses-15-01380-t001:** Summary of the most significant viral genomes assembled from high-throughput sequencing.

Virus	Contig Length (nt)	Genomic RNA Segment	Number of Reads Assembled into Contig (%) ^(1)^	Average Sequencing Depth (per nt)	GenBank Closest Hit (Accession Number)	Nucleotide Identity (%) (Query Coverage%) to Closest Hit
Babaco virus Q	4707	-	6,534,602 (11.940%)	144,256	Babaco virus Q (MN648673)	98% (100%)
New babaco ilarvirus	3201	RNA1	15,806 (0.028%)	482.5	Apple necrotic mosaic virus, RNA1 (KY808376)	68% (60%)
	2508	RNA2	4221 (0.007%)	153	Apple necrotic mosaic virus, RNA2 (LC108994)	71% (53%)
	1719	RNA3	13,788 (0.025%)	760	Prunus necrotic ringspot virus, RNA3 (MZ451064)	67% (36%)
New babaco rhabdovirus	12,770	-	3633 (0.006%)	32.3	Joa yellow blotch-associated virus (MW014292)	68% (91%)
Endogenous pararetrovirus (BabEPV)	632–2682	-	3681 (0.007)	4–108	Citrus endogenous pararetrovirus (ARO38276)	89% (45%)
Babaco cryptic virus (BabCV)	360–382	-	34 (0.000062)	3	Babaco cryptic virus—1 (MN648674)	98% (87%)

^(1)^ From total reads post-trimming (54,689,746).

**Table 2 viruses-15-01380-t002:** Amino acid identities (%) between hypothetical proteins of babaco ilarvirus-1 (BabIV-1) and counterparts from closest ilarviruses. The accession numbers for the protein 1a (P1a), RNA-dependent-RNA-polymerase (RdRp), movement protein (MP), and the coat protein (CP) of subgroup 3 ilarviruses are listed in Appendix A.

Ilarviruses	P1a	RdRp	MP	CP
Subgroup 3				
Apple mosaic virus	73.96	65.63	71.33	61.61
Apple necrotic mosaic virus	74.46	71.98	72.79	70.00
Blueberry shock virus	74.53	66.16	73.58	68.56
Prunus necrotic ringspot virus	76.05	67.73	73.61	68.72
Lilac leaf chlorosis virus	78.24	64.97	81.79	72.85
Average (% identity)	75.45	67.29	74.62	68.35

**Table 3 viruses-15-01380-t003:** Amino acid identities (%) between the hypothetical proteins of babaco rhabdovirus-1 (BabRV-1) and counterparts from closest alphanucleorhabdoviruses. The accession numbers for the nucleocapsid (N), accessory ‘X’ protein, phosphoprotein (P), movement protein (MP), matrix (M), glycoprotein (G), and polymerase (L) are listed in Appendix A.

	Amino Acid Identity (%)						
Virus	N	X	P	MP	M	G	L
Joa yellow blotch-associated virus	70.76	28.95	58.06	67.13	61.87	66.19	74.36
Potato yellow dwarf virus	71.52	27.78	52.50	67.94	68.87	66.40	73.75
Physostegia chlorotic mottle virus	43.15	32.69	20.47	42.16	35.27	39.24	53.32
Eggplant mottled dwarf virus	41.70	25.47	18.52	42.16	32.56	37.69	52.91
Peach virus 1	24.17	Not applicable	15.51	10.19	14.60	21.83	33.25

## Data Availability

Virus genomic sequences generated in this work have been deposited in the GenBank of the National Center for Biotechnology Information (NCBI) under accession numbers OQ256238, OQ256239, OQ256240, and OQ256237.

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
