# Peer review of "The Virome of Babaco (Vasconcellea × heilbornii) Expands to Include New Members of the Rhabdoviridae and Bromoviridae"

_viruses, 2023, doi:10.3390/v15061380_

Round 1

Reviewer 1 Report

Review report on the manuscript of Reyes-Proaño entitled: „The virome of babaco (Vasconcellea x heilbornii) expands to include new members of the Rhabdoviridae and Bromoviridae”

 In this manuscript the authors investigated the causative agent of a symptomatic babaco using traditional methods and HTS. Using RT-PCR they revealed that the plant is infected with several viruses: BabVQ, BabCV and BabEPV, but these are known to be latent viruses why they further investigated the plant using ribodepleted RNAseq. As a result of the study the presence of an Ilarviruss and an alphanucleorhabdovirus was identified. Both three RNAs of the Ilarvirus and the entire alphanucleorhabdovirus were amplified in several RT-PCR reactions and sequenced, showing that they are new viral species. The genomes were completed using 5’ and 3’ RACE and it phylogenetic relationship were also analysed.

Biotest using different plant failed using papaya, babaco and chamburo. Result of a small survey shown infection with the new viruses either alone or as a coinfection, but their presence did not coincide with any symptom.

I think this precise work is a very nice description of two new viruses and can raise the attention of the neglected sanitation of babaco, a vegetatively propagated plant.

I only have some minor suggestions and recommendation to the authors. After a small revision I think this manuscript can be accepted in MDPI Viruses.

Minor comments: (page numbering starts only at page 9, in my comments I used the counted page numbers)

1/ in the abstract I won’t cite numbers (nt of the genome, % identity), but phrase the finding on a more generalized way.

2/ In the begining of the abstract I would write what was the aim of the research and what methods were used.

3/page3 line 2 I would add what methods was used to test the presence of already known viruses and consider to add the used diagnostic primers to the STable 1 with their citation.

4/page4, line 9. I guess not whole genome sequencing rather ribodepleted RNA seq was the used method

5/page5 line20, please clarify the letter symbols

6/page5 line 06: i would add …leaves of the original virus infected plants were…

7/ I would include read data for all viruses (including the cryptic viruses

8/page 7 and 11: I would start the genome organization chapter with a short paragraph about how the genomes were completed and sequenced. It is detailed in the materials and methods section, but would be nice to summarize here

9/Figure2: colour of the helicase and methyltransferase are mixed up in the legend and on the figure itself

10/ referring to pfam motifs are not correct at several places – instead of pfam, PF should be used to be able to find the motif in the database

11/ nucleotide identities would be better to shown in a similar was for both viruses. If the style of Table 3 (simple) is preferred it can be leave in the main text. If the detailed style of Table 2 is preferred I would suggest to consider to put it into the supplementary.

12/ I would also consider to keep the Table 4 for the supplementary

13/ I would include a table about the survey results, including cultivar or variety info if there is, or possible symptoms – if there is, about the infected plants. If this would be a complex table it could be placed into the supplementary.

14/ please consider to rephrase the sentence of page 16 line 206, to suggest that the vegetative propagation is the worst and not the best practice. In this form although it is true it could be misread.

 As a summary I think this is a nice work and after some minor correction it can be accepted in MDPI Viruses for publication.

Reviewer 2 Report

In the current manuscript, the authors have detected two novel viruses closely related to ilarvirus genus (Bromoviridae), and alphanucleorhabdovirus (Rhabdoviridae) infecting babaco plant in Ecuador. The authors utilized high throughput sequencing (HTS) approach to identify virome of a symptomatic plant. Following bioinformatic analyses, PCR confirmation of the contigs, and phylogeny analyses, they are reporting two novel viruses with provisional names, babaco ilarvirus 1 (BabIV-1), and babaco nucleorhabdovirus 1 (BabRV-1). They have also performed a small survey of the identified viruses, and found out a distribution of BabIV-1 in 21% and BabRV-1 in 36% of the collected samples. The study is informative but it needs to address some major and minor comments specially in methodologies and results sections prior to recommendation for publication in Viruses. Please find the comments below:

1- The page numbers is not organized, and this made it difficult to track the review report.

1- The abstract lacks a summary of the followings: a- background information on plant and its importance and why the study is done on such plant species. b- methodologies used, whether molecular, bioinformatic, or bioindexing.

2- line 46 in introduction, what is "family level" ? rephrase the word.

3- line 7 second page of introduction, add reference for viruses detected.

4- how old was the plant used in the study? describe in the materials and methods.

5- line 9 section "virus sequencing, assembly, and annotation", please revise the phrase to explain plant RNA was subjected to HTS.

6- line 17 the same section as above, add explanation to the manuscript why using another plant as reference for bioinformatic analyses.

7- I am not sure if the authors have done the RACE-PCR; I don't see any methodologies explained. If it hasn't been done, this must be done to report the complete genome of each virus.

8- I didn't find the accession numbers reported from NCBI, please provide a temporary access to the reviewers.

9- in figure 2, remove the bottom scale ranging from 0 thru 3500; it is redundant and doesn't contain any information. Also, what are the gray color parts spanning each ORF; add explanation as the caption.

10- table 2 is best to be as a supplementary table; it is confusing if placed in the main content of the paper.

11- figure 3, please use the 'whole genome' of novel ilarvirus for phylogenetic analyses rather than the individual genes. This way, it gives a broader idea of genetic information of the novel virus with its close species. Use the single phylograms as a supplementary figure.

12- table 3, needs to be a supplementary table.

13- table 4, use a software such as T-Coffee to provide an easy-to-observe alignment.

14- figure 4, add legend to each ORF such as the ones in figure 2, add nucleotide or amino acid sizes. Make sure the right sequence ending in "AAAAGC" doesn't have dots at its end.

15- figure 5, describe the "asterisk" sign in the caption.

16- line 151, please provide the observation/results on the mechanical inoculation experiments. I highly recommend the authors to include bioindexing on typical plants such as quinoa or tobacco, since they have been widely used to provide a quick information about virus biology.

17- line 154, I don't see the methodologies used for the survey, such as how old were the plants sampled, how many leaves were collected per plant, were the plants symptomatic or not, how the leaves were processed, what molecular method, primers and conditions used,

18- line 179, this phrase is inconclusive due to lack of proof in the study. Additionally, it could be due to coinfection with other viruses reported in the study.

19- lines 193-194, there is no proof of individual viral infection in this study to rule out the coinfection impact on symptom appearance. Therefore, this sentence is inconclusive.

20- the journal names in the references should be unanimous and followed by the Viruses guideline; some are abbreviated and some are complete.

Some phrases are too long; while keeping the sentences short, they should be informative as well. So, please rephrases the sections below:

1-abstract, line 28.

2-results, line 43 "virus detection"-related paragraph.

Reviewer 3 Report

The manuscript by Reyes-Proaño et al. describes the characterization of two new viruses, BabIV-1 and BabRV-1 infecting babaco. The work is well conducted and the results represent an important contribution to the knowledge of the virome of this crop. Authors detect and characterize the virus genomes by HTS and confirm infection status by RT-PCR. 

In my opinion there is a weakness in this very nice work concerning the impact of the viruses in producing areas, as the tested samples are all from a nursery. I suggest authors, in order to improve the significance of the work, to perform a small survey in different growing areas, which would provide a better picture of the incidence of these pathogens.

Another important point is related to data availability. As sequences submitted to NBCI are not available yet, I would like authors to submit recovered genomes for the revision process. Also, according to MDPI data availability policy, if the paper is finally accepted, authors should provide HTS reads supporting their findings, at least upon request .

In addition, I have the following suggestions:  

Material and methods: 

1)    Symptomatology description is a result, this information as well as figure 1 should be moved to results section. 

Related to this, in material and methods, information on the 50 plants analyzed by RT-PCR.

2)    In addition of providing a reference for dsRNA extraction protocol, authors should describe the procedure.

3)    Authors should revise carefully and complete reagent supplier affiliations. 

4)    Although listed in table S1, authors should include in the manuscript primers developed for detection.

Results:

1)    Page 7, lines 2-3. The fact that plants tested negative (positive) for several viruses does not suggest the absence (presence) of additional viruses. This part of the sentence should be removed. 

2)    Figure 3 caption is too long. I recommend authors to include some of this information, for example virus names, in material and methods and add only abbreviation in figure 3 caption.

3)    Figure 4B. Please check, I do not follow well trailer sequence in the proposed secondary structure. Trailer seems to be split and two nucleotides (UU) at 5-6 positions from 5’ end are missing. In addition, as far as I can see the sequence is written in the wrong sense.

4)    Figure 5. Caption is too long. Transfer information to material and methods 

Discussion:

I have a major concern on the authors conclusions related to symptomatology. The statement that “BabIV-1 could be the sole causal agent for this symptom” is not supported at all by the data presented in the paper, specially taking into account the reduced number of plants analyzed. This sentence should be removed from the discussion.

Moreover, in the discussion authors seem to suggest (although not clearly stating it) that rhabdoviruses do not produce severe symptomatology, which is not correct. I suggest authors to revise this section to clarify this point. 

Other minor points:

Page 2, lines 27-28. Rewrite “The tripartite genome of the new ilarvirus, provisionally named babaco ilarvirus 1 (BabIV-1), consisted of...”

Page 2, line 29. Start a new sentence: “BabIV-1 is more closely related to subgroup ilarviruses….”

Page 2, line 32. Change “consisting of” to “consisted of”

Page 2, lines 46-49. What do authors mean with “nursery operations”? If they mean “agronomic practices” which can contribute to virus dissemination I suggest to rewrite “Nevertheless, the vegetative propagation, the lack of pathogen-free certified material as well as nursery operations have caused an exponential dissemination of viruses in the main producing areas threatening the…”

Page 2, line 49. Change “production” to “producing”

Page 3, line 12. Add “family” in both brackets 

Page 3, line14 and line 27. Add references of ICTV reports.

Page 3, line 37. Change “which” to “and”

Page 4, line 17. Remove “the”

Page 4, line 18. Change “RT” to “RT-PCR”

Page 4, lines 13-23. Authors should include specific primers sequence

Page6, line 38. Remove “operation”

Page7, line 18. Change “virus-associated” to “virus-related”

Page 8, line 11. Remove error message

Page 8, line 19. Remove error message

Pag 14, line 112. Change “Alphanucleorhabdovirus genus” to “genus Alphanucleorhabdovirus”

Page 15, line 120. Give reference for the ICTV report

Page 15, line 123. Rewrite “solanaceous” and remove italics

Page 15, line 137. “CBDaV” remove underline. 

Page 16, line 170. Change “Ilarvirus genus” to “genus Ilarvirus”

Minor editing of English required

Round 2

Reviewer 2 Report

I thank the authors to address the comments I provided.

Regarding figure 3 for Ilarvirus phylogenetic analysis, since each genomic segment is closely related to similar viruses, I believe there shouldn't be a problem in retrieving a whole genome basis phylogram. If the result is close to individual phylograms, please add it to figure 3.

Since typical bio indexing plants (quinoa, tobacco) were not used in the inoculation study and the authors would like to pursue with the infectious clones, please provide information in the discussion section related to future research using infectious clones to determine viral pathogenicity.

Author Response

Response to Reviewer:

Thank you very much for your time. Next we address each of the two observations made by the reviewer.

Reviewer: Regarding figure 3 for Ilarvirus phylogenetic analysis, since each genomic segment is closely related to similar viruses, I believe there shouldn't be a problem in retrieving a whole genome basis phylogram. If the result is close to individual phylograms, please add it to figure 3.

Response:

We kindly and respectfully disagree one more time on this suggestion.

Nucleotide identities between the new babaco ilarvirus and closest relatives are very low and do not provide a reliable (consistent bootstrap values) phylogenetic topology. Ilarviruses are segmented (tri-partite) RNA viruses that must be analyzed by segments. Our study does not involve a population structure component, where several isolates of the same virus are compared, in which case complete genome analyses might be suitable. Our analyses, at the amino acid level, of each protein provides a reliable phylogram altogether.

Reviewer: Since typical bio indexing plants (quinoa, tobacco) were not used in the inoculation study and the authors would like to pursue with the infectious clones, please provide information in the discussion section related to future research using infectious clones to determine viral pathogenicity.

Thank you. We have added a few lines to the next-to-the-last paragraph in the discussion to satisfy this suggestion.